# A multistate assessment of population normalization factors for wastewater-based epidemiology of COVID-19

**Andrew L. Rainey** [1,2], **Song Liang** [1,2], **Joseph H. Bisesi, Jr.** [1,2,3], **Tara Sabo-Attwood** [1,2,3], **Anthony T. Maurelli** [1,2] *

**1** Department of Environmental and Global Health, College of Public Health and Health Professions, University of Florida, Gainesville, Florida, United States of America, **2** Emerging Pathogens Institute, University of Florida, Gainesville, Florida, United States of America, **3** Center for Environmental and Human Toxicology, University of Florida, Gainesville, Florida, United States of America

☯ These authors contributed equally to this work.
\* amaurelli@phhp.ufl.edu

## Abstract

Wastewater-based epidemiology (WBE) has become a valuable tool for monitoring SARS-CoV-2 infection trends throughout the COVID-19 pandemic. Population biomarkers that measure the relative human fecal contribution to normalize SARS-CoV-2 wastewater concentrations are needed for improved analysis and interpretation of community infection trends. The Centers for Disease Control and Prevention National Wastewater Surveillance System (CDC NWSS) recommends using the wastewater flow rate or human fecal indicators as population normalization factors. However, there is no consensus on which normalization factor performs best. In this study, we provided the first multistate assessment of the effects of flow rate and human fecal indicators (crAssphage, F+ Coliphage, and PMMoV) on the correlation of SARS-CoV-2 wastewater concentrations and COVID-19 cases using the CDC NWSS dataset of 182 communities across six U.S. states. Flow normalized SARS-CoV-2 wastewater concentrations produced the strongest correlation with COVID-19 cases. The correlation from the three human fecal indicators were significantly lower than flow rate. Additionally, using reverse transcription droplet digital polymerase chain reaction (RT-ddPCR) significantly improved correlation values over samples that were analyzed with real-time reverse transcription quantitative polymerase chain reaction (rRT-qPCR). Our assessment shows that utilizing flow normalization with RT-ddPCR generate the strongest correlation between SARS-CoV-2 wastewater concentrations and COVID-19 cases.

## Introduction

The Coronavirus disease 2019 (COVID-19) pandemic has challenged public health officials conducting geographically widespread and comprehensive disease surveillance throughout communities. There was an urgent need to develop new strategies to generate real-time data to monitor trends in infections, identify potential outbreaks, and detect the emergence of new

**Data Availability Statement:** CDC NWSS data that was used for this study is publicly available and can be found on their website (https://data.cdc.gov/Public-Health-Surveillance/NWSS-Public-

SARS-CoV-2-Wastewater-Metric-Data/2ew6-ywp6).

**Funding:** ALR National Center for Advancing Translational Sciences of the National Institutes of Health under University of Florida and Florida State University Clinical and Translational Science Awards TL1TR001428 and UL1TR001427 (https://ncats.nih.gov/) The funders had no role in study design, data collection and analysis, decision to publish, or preparation of the manuscript.

**Competing interests:** The authors have declared that no competing interests exist.

severe acute respiratory syndrome coronavirus 2 (SARS-CoV-2) variants [1,2]. Early in the pandemic, wastewater-based epidemiology (WBE) emerged as a rapid, cost-effective approach that overcomes limitations of traditional disease monitoring methods (e.g., their dependence on individual health seeking behaviors) to generate much-needed surveillance data [3–6]. WBE was a viable option because, though SARS-CoV-2 is a respiratory pathogen, the virus also replicates in the human intestine and is shed in the feces of an infected individual, irrespective of their symptomatic status [7–9]. In September 2020, the Centers for Disease Control and Prevention (CDC) developed the National Wastewater Surveillance System (NWSS) to support and coordinate SARS-CoV-2 wastewater surveillance programs across the United States (U.S.) [6]. WBE of SARS-CoV-2 has been extremely successful in monitoring infection trends among a broad range of communities across the globe [10–15]. Measurement of SARS-CoV-2 concentrations in wastewater to monitor trends in infections is a key analytical approach of SARS-CoV-2 WBE. The data generated by SARS-CoV-2 WBE issued to inform key stakeholders making decisions such as utilization of mobile test clinics, allocation of clinical testing resources, or changing social distancing and mask wearing requirements in public spaces [6]. However, using raw SARS-CoV-2 wastewater concentrations to monitor infection trends does not account for the relative human contribution to a wastewater sample that is collected for analysis. This is important because a wastewater sample collected for analysis may not have fecal input from the entire human population within a community, or it may have additional non-human fecal contributions from animals that live in the surrounding environment. Adjusting SARS-CoV-2 wastewater concentrations to account for the relative human contribution may improve the data for analysis and interpretation.

Normalization of wastewater concentrations with a population biomarker has been long studied, however, there is no consensus on a single normalization factor that is best for WBE of SARS-CoV-2. Population normalization factors can be endogenous or exogenous physical, chemical, or biological biomarkers that are analyzed and quantified from a wastewater sample [16–19]. Examples of previously used population normalization factors used for SARS-CoV-2 WBE are wastewater flow rate, human fecal indicator organisms, electrical conductivity, ammonium, caffeine, paraxanthine, creatinine, and 5-hydroxyindoleacetic acid [20–25]. Currently, CDC NWSS recommends wastewater flow rate and human fecal indicator organisms for normalizing SARS-CoV-2 wastewater concentrations for disease trend analysis [26]. Human fecal indicators are bacteria and viruses that are shed exclusively in human feces and are measured to quantify the relative human population that are represented within a wastewater sample. Human fecal indicators used in SARS-CoV-2 WBE include Pepper Mild Mottle virus (PMMoV), crAssphage, Bacteroides HF183, or F+ Coliphage [22,23,27,28]. Previously, human fecal indicators have been shown to display mixed associations (yes and no association) with estimated serviced populations from wastewater catchment areas [22,29,30]. Studies that measured the ability of human fecal indicators to improve SARS-CoV-2 WBE analysis are also inconclusive, as they show positive or no positive effect on the correlation between SARS-CoV-2 wastewater concentrations and COVID-19 clinical cases [23,27,29].

The objective of this study is to assess the CDC NWSS recommended population normalization factors to determine which is best for monitoring COVID-19 trends within a community. We analyzed a large, multi-state CDC NWSS dataset to identify which population normalization factor provides the strongest correlation between SARS-CoV-2 wastewater concentrations and reported COVID-19 cases. This study has provided empirical evidence that the optimal analytical approach for SARS-CoV-2 WBE is to use the wastewater flow rate as a population normalization factor. We recommend that wastewater flow also be considered for population normalization in wastewater surveillance of other infectious diseases.

## Methods

### Data source and study sites

We obtained data which includes 18 months of data from six study sites, i.e., U.S. states that contributed SARS-CoV-2 wastewater data to CDC NWSS to help track and monitor COVID-19 in communities across the country [31]. The study sites include the following U.S. states: California (CA), Colorado (CO), North Carolina (NC), Ohio (OH), Virginia (VA), and Wisconsin (WI). Each study site provided wastewater data collected from multiple different sewersheds, which are defined as a geographic area that contributes wastewater to a specified point in the wastewater collection system (e.g., to a treatment plant or other sample collection location). The estimated service population of the individual sewershed was provided to CDC NWSS from each study site. Most sewersheds service an entire city, while some sewersheds service a smaller portion of a single city. Influent wastewater collected from each sewershed was used for sample analysis and data reporting to NWSS. These six study sites were included for our analysis based on availability of the following data variables: SARS-CoV-2 raw wastewater measurements, influent wastewater flow rate, fecal indicator raw wastewater measurements, and sewershed-level COVID-19 clinical cases. In total, 9,705 wastewater samples were analyzed across 182 sewersheds in the six study sites from May 27, 2020—October 4, 2021. Descriptive statistics of the study sites are summarized in Table 1.

### COVID-19 clinical cases

The daily confirmed COVID-19 cases at the sewershed level were reported to CDC NWSS by each reporting study site. A COVID-19 clinical case is defined by the previously described CDC COVID-19 case definition [32]. The COVID-19 case definition includes criteria that must be met to classify an individual as a suspected, probable, or confirmed COVID-19 case. COVID-19 clinical case data in this analysis only include probable and confirmed cases as classified by CDC [32]. The COVID-19 clinical cases corresponded to each individual wastewater sampling date for a given sewershed. To protect individual privacy, the daily number of COVID-19 cases for a given sewershed were suppressed if the total number of cases was greater than zero but less than five. For our analysis, we assigned a value of two COVID-19 cases to any observation that was suppressed.

### Wastewater processing methods

Each lab that processed wastewater samples included the concentration and extraction procedures used to process the wastewater samples for molecular analysis. The specific concentration and extraction methods used by each study site are found in S1 Table. In general, concentration methods included ultracentrifugation, ultrafiltration, or electronegative membrane filtration. Extraction of nucleic acid was carried out using either column- and magnetic

**Table 1. Descriptive statistics of the CDC NWSS study sites included for analysis.**

| Study Site | Sampling Dates | Number of Sewersheds | Number of Observations | Estimated Sewershed-Level Population Served |
|---|---|---|---|---|
| North Carolina | December 13, 2020—October 4, 2021 | 22 | 989 | 3,500–550,000 |
| Wisconsin | August 23, 2020—October 4, 2021 | 60 | 3,801 | 3,100–615,934 |
| Colorado | August 2, 2020—September 30, 2021 | 20 | 1,914 | 5,818–709,904 |
| Virginia | May 27, 2020—September 21, 2021 | 9 | 623 | 69,059–343,016 |
| California | September 7, 2020—October 3, 2021 | 10 | 1,234 | 40,000–4,000,000 |
| Ohio | July 26, 2020—September 27, 2021 | 61 | 3,884 | 3,300–654,817 |

bead-based extraction kits, and also the 4S method was used by one processing lab [33]. Each study site used a SARS-CoV-2 surrogate as a process control for each wastewater sample that ensured there was no error in wastewater concentration, nucleic acid extraction, or nucleic acid amplification. Surrogates used as process controls included Human Coronavirus OC-43, Bovine Coronavirus, Murine Coronavirus, and Hepatitis G.

We included SARS-CoV-2 wastewater measurements of the CDC N1 and/or CDC N2 genetic targets [34]. Virginia included SARS-CoV-2 wastewater measurements using the ddCoV_N genetic target that is located on the same region as the CDC N2 genetic target, thus it was included as an N2 wastewater measurements for our analysis [35]. SARS-CoV-2 wastewater measurements were performed using either real-time reverse transcription quantitative polymerase chain reaction (rRT-qPCR) or reverse transcription droplet digital polymerase chain reaction (RT-ddPCR). California and Wisconsin used both molecular analysis methods, while the remaining study sites exclusively used RT-ddPCR. The molecular assay limit of detection was provided for each sample by the reporting laboratory. For our analysis, we assigned a value of half of the limit of detection for SARS-CoV-2 negative wastewater samples. All SARS-CoV-2 wastewater measurements were initially reported as the number of genomic copies per liter of wastewater (GC/L), which we converted to $Log_{10}GC/L$ for statistical analysis.

## Population normalization factors

We normalized the wastewater concentrations using the fecal indicator wastewater concentration and/or the flow rate for each SARS-CoV-2 wastewater observation. PMMoV, crAssphage, and F+ coliphage were the fecal indicator organism targets used by the included study sites. PMMoV was analyzed across every study site, crAssphage was analyzed in a subsample of wastewater collected in Ohio, and F+ coliphage was analyzed in a subsample of wastewater collected in Colorado. The fecal indicator organism nucleic acid was concentrated extracted and analyzed using the same methods as used for SARS-CoV-2 nucleic acid (S1 Table). No limit of detection of the molecular assay for the fecal indicator organisms was provided, therefore, we did not assign numeric values to any negative/missing fecal indicator measurements. The fecal indicator wastewater concentration was reported as GC/L and we converted this value to $Log_{10}GC/L$ for statistical analysis. We performed fecal indicator normalization of the SARS-CoV-2 concentrations in wastewater by dividing the SARS-CoV-2 wastewater concentration ($Log_{10}GC/Day$) over each fecal indicator wastewater concentration ($Log_{10}GC/Day$) to obtain a unitless ratio of the two values, i.e., SARS-CoV-2 concentration to fecal indicator concentration. We also conducted fecal normalization with the flow normalized SARS-CoV-2 and fecal indicator wastewater concentrations ($Log_{10}GC/Day$) to obtain the unitless flow and fecal normalized value.

The flow rate for each observation was reported by each sewershed for the specific day the raw influent wastewater sample was originally collected. The flow rate was reported as million gallons per day (MGD), and we converted this value to liters per day (L/Day) for our analysis. To flow normalize, we multiplied the initial SARS-CoV-2 wastewater measurement (GC/L) with the flow rate (L/Day) to obtain the final value of genomic copies per day (GC/Day), which we then converted to $Log_{10}GC/Day$ for statistical analysis. We also conducted fecal normalization (described above) with the flow normalized SARS-CoV-2 and fecal indicator wastewater concentrations ($Log_{10}GC/Day$) to obtain the unitless flow and fecal normalized value.

## Statistical analysis

We computed the descriptive statistics such as the mean, and ranges of the COVID-19 clinical cases and the measured wastewater concentrations of SARS-CoV-2 and the fecal indicators.

The descriptive statistics of the COVID-19 clinical cases were computed using the total daily sum of the sewershed-level cases across all sewersheds within each study site.

We analyzed the correlation between the raw CDC N1 and CDC N2 wastewater concentration ($Log_{10}GC/L$) of individual wastewater samples to determine the agreement of the quantified measurement of each genetic target. This analysis was then stratified by the different molecular analysis method used (rRT-qPCR and RT-ddPCR). This analysis was not conducted for observations from the state of Colorado because only the CDC N1 genetic target was used to analyze wastewater samples from this site. The correlation between the raw fecal indicator wastewater concentration ($Log_{10}GC/L$) and the flow rate (L/Day) with the estimated population served by each sewershed was analyzed to determine if any normalization parameters were accurate population markers.

The next correlation analysis was performed between the SARS-CoV-2 wastewater concentrations with the corresponding sewershed-level COVID-19 clinical cases of each wastewater observation. The correlation analysis was conducted among the following four normalization categories: the raw SARS-CoV-2 wastewater concentration ($Log_{10}GC/L$), the flow normalized SARS-CoV-2 wastewater concentration ($Log_{10}GC/Day$), the raw fecal normalized value (for each individual fecal indicator), and the flow and fecal normalized value (for each individual fecal indicator). This test was performed for each study site, aggregating observations across all the sewersheds included within a study site, i.e., a U.S. state. This correlation analysis was stratified by the genetic targets (CDC N1 and CDC N2) of the molecular assay. The correlation analysis was also stratified by the molecular analysis method used, i.e., rRT-qPCR and RT-ddPCR. Multiple t-tests were performed using the range of the stratified correlation coefficients observed from the analysis of the 1) molecular analysis method used (rRT-qPCR and RT-ddPCR), 2) the SARS-CoV-2 genetic target (CDC N1 and N2) measured, and 3) across the normalization categories (raw, flow normalized, fecal normalized, flow and fecal normalized). All correlation analysis results were generated using the Spearman's correlation coefficient. All statistical tests used a P-Value significance level of $\alpha = 0.05$. All statistical analyses were conducted using RStudio version 2022.07.01 [36,37]. Figures were generated using the ggplot2 package in RStudio [38].

## Results and discussion

### COVID-19 clinical cases

The range and mean value of the daily sewershed-level COVID-19 clinical cases reported across the study sites are summarized in Table 2. Overall, daily sewershed-level COVID-19 clinical cases ranged from 2–5,805 cases, and the mean value from 24–209, with the highest mean value in California. The date with the highest number of reported COVID-19 cases (5,805 cases) was January 4, 2021, in California, in a community with an estimated 3.5 million individuals served by the corresponding sewershed. The timing of this high volume of cases could be representative of the surge of new COVID-19 cases associated with the highly contagious SARS-CoV-2 Omicron variant [39]. Overall, the COVID-19 pandemic had a significant burden of disease among our study sites throughout the study period.

### SARS-CoV-2 wastewater concentrations

The raw SARS-CoV-2 wastewater concentration ($Log_{10}GC/L$) for the CDC N1 and N2 genetic targets across the study sites are found in Table 2. Overall, the SARS-CoV-2 wastewater concentration ranged from 3.64–7.94 $Log_{10}GC/L$ (N1) and 3.91–7.14 $Log_{10}GC/L$ (N2). The lowest quantified SARS-CoV-2 wastewater concentration for the N1 genetic target was observed in Virginia (3.64 $Log_{10}GC/L$) and the lowest N2 genetic target wastewater concentration was

**Table 2. Descriptive statistics of the sewershed-level COVID-19 clinical cases, SARS-CoV-2 raw wastewater concentration (Log$_{10}$GC/L), and fecal indicator raw wastewater concentration (Log$_{10}$GC/L).**

| State | Sewershed-Level Daily COVID-19 Case Range (Mean) | CDC N1 Wastewater Mean Concentration (±Standard Deviation) | CDC N2 Wastewater Mean Concentration (±Standard Deviation) | Fecal Indicator Wastewater Mean Concentration (±Standard Deviation) | | |
|---|---|---|---|---|---|---|
| | | | | PMMoV | F + Coliphage | crAssphage |
| **North Carolina** | 2–586 (35) | 3.90 (±0.74) | 3.91 (±0.89) | 7.61 (±0.37) | NR | NR |
| **Wisconsin** | 2–605 (24) | 4.72 (±0.63) | 4.60 (±0.70) | 7.23 (±0.40) | NR | NR |
| **Colorado** | 2–1,651 (59) | 4.61 (±0.52) | NR | 6.55 (±0.34) | 7.82 (±0.31) | NR |
| **Virginia** | 2–290 (35) | 3.64 (±0.57) | 3.98 (±0.61) | 7.55 (±0.45) | NR | NR |
| **California** | 5–5,805 (209) | 4.72 (±0.83) | 4.72 (±0.77) | 7.77 (±0.64) | NR | NR |
| **Ohio** | 2–2,854 (29) | 4.00 (±0.84) | 4.01 (±0.87) | 6.98 (±0.57) | NR | 7.36 (±0.85) |

NR = Not Reported.

observed in North Carolina (3.91 Log$_{10}$GC/L). The highest SARS-CoV-2 wastewater concentration for both, N1 (7.94 Log$_{10}$GC/L) and N2 (7.14 Log$_{10}$GC/L), was observed in California (Table 2).

It is both common and recommended to analyze wastewater nucleic acid extracts for more than one SARS-CoV-2 genetic target [11,40,41]. Many studies have investigated differences in the SARS-CoV-2 wastewater concentration measured by the CDC N1 and N2 genetic targets and have largely been inconclusive as to whether one target is more effective than the other [42–45]. We analyzed the N1 and N2 wastewater concentrations across each study site to determine the level of agreement between quantified measurements for each genetic target from an individual wastewater observation. The Spearman correlation coefficients of the N1 and N2 genetic target measurements among each study site are found in Table 3. The strongest correlation was in the state of North Carolina (R = 0.93, P-Value<0.001) and the lowest was in Ohio (R = -0.06, P-Value = 0.23). However, it should be noted that in Ohio, <10% of all wastewater samples were analyzed for both N1 and N2. No correlation was computed for Colorado because wastewater samples were only measured using the N1 genetic target. Our results of the correlation of the N1 and N2 genetic target wastewater concentrations were high (Table 3), and consistent with data reported from other published studies across the globe [10,23,43,46].

**Table 3. Spearman correlation coefficient analysis of the measured CDC N1 and CDC N2 wastewater concentration (Log$_{10}$GC/L) of single wastewater samples across each sample site.**

| Study Site | Correlation Coefficient (P-Value) | | |
|---|---|---|---|
| | Overall | RT-ddPCR | rRT-qPCR |
| **North Carolina** | 0.93 (<0.001) | 0.93 (<0.001) | NR |
| **Wisconsin** | 0.90 (<0.001) | 0.92 (<0.001) | 0.87 (<0.001) |
| **Virginia** | 0.89 (<0.001) | 0.89 (<0.001) | NR |
| **California** | 0.88 (<0.001) | 0.84 (<0.001) | 0.72 (<0.001) |
| **Ohio** | -0.06 (0.23) | NR | -0.06 (0.23) |

NR = Not Reported.

## Association of normalization factors with estimated population served

The first step to identifying the optimal population normalization factor is to identify which wastewater biomarker is most strongly associated with the estimated service population of a sewershed. This estimate was provided by each sewershed, and remained at a static value that did not adjust for real-time fluctuations in the serviced populations that may be influenced by factors such as travel by individuals for work or holidays. It is currently recommended by CDC NWSS to use either wastewater flow rate or a human fecal indicator organism as population normalization factors. We investigated flow rate and three recommended fecal indicator biomarkers to determine which are most associated with the corresponding estimated service population of a sewershed. The range of the correlation coefficients for the fecal indicator wastewater concentrations and the flow rate with the estimated population served can be found in Fig 1. A T-Test of the overall correlations stratified by the population biomarker showed that the correlations between the flow rate and estimated population served were significantly higher than the correlations of the fecal indicator concentrations (Mean difference = 0.68, P-value<0.001), and also significantly higher than the flow normalized fecal indicator concentrations (Mean difference = 0.19, P-value<0.001) with the estimated population served (Fig 1). The correlation of the flow normalized fecal indicators and the estimated population served was also significantly higher than the correlation of the fecal indicator concentrations and the estimated population served (Mean difference = 0.49, P-value = 0.033) (Fig 1). The individual correlation coefficients of the fecal indicators and the flow rate with the estimated population served across each sewershed can be found in S2 Table. Overall, the correlations of the raw wastewater concentrations ($Log_{10}GC/L$) for all three human fecal indicator organisms (crAssphage, F+ Coliphage, PMMoV) with the estimated service population were low to moderate. The strongest correlation among the fecal indicators were observed with PMMoV in North Carolina, Wisconsin, Colorado, and California. The strongest correlation being in California using PMMoV (R = 0.56). There was no significant correlation observed for PMMoV in Ohio (R = 0.03, P-Value = 0.41) and Virginia (R = -0.04, P-Value = 0.47). The

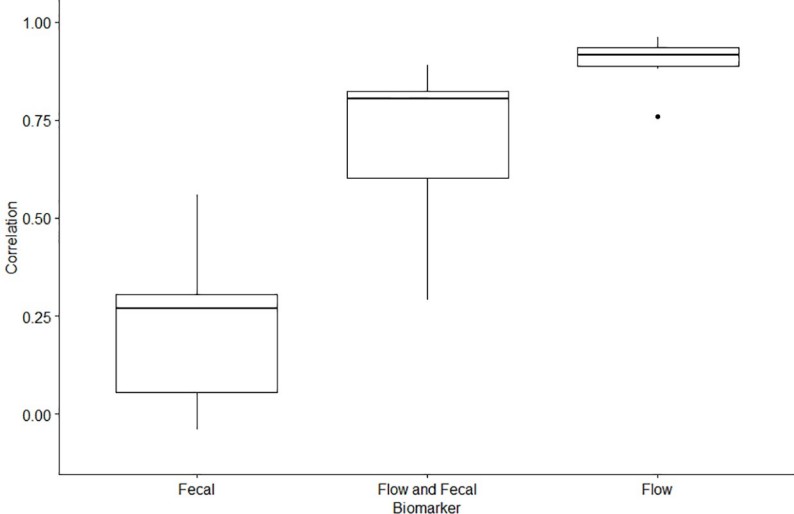

**Fig 1. Correlation coefficients of the fecal indicator raw wastewater concentration ($Log_{10}GC/L$), fecal indicator flow normalized wastewater concentration ($Log_{10}GC/Day$), and flow rate (L/Day) with the estimated population served across each study site.** Fecal indicator correlations include analysis results of PMMoV, crAssphage, and F + Coliphage.

moderate correlation observed for PMMoV and the populations in our study was consistent with a study from Missouri which reported a similar correlation (R = 0.36) for PMMoV with the population served in their study [21]. Other studies have also reported no correlation with the PMMoV concentration and the population served [22,47]. Another study in Minnesota also reported a low to moderate association between the population served and PMMoV wastewater measurements [27]. One explanation for these results is that PMMoV is a plant virus, and the presence and concentration of PMMoV in human feces is strongly dependent on the dietary habits of the individuals within a community [48]. While PMMoV has been detected in wastewater across the globe without seasonal variability, it has been shown to have some geographic variability, likely due to regional diets, which can be a limitation when trying to use PMMoV wastewater concentrations as a population biomarker [49,50]. The discrepant correlations of PMMoV wastewater concentrations with populations served in our study and others means that more research should be conducted before PMMoV can be considered as a suitable population biomarker for WBE. The crAssphage fecal indicator target, which was only used in a subset of data from Ohio (20 sewersheds, 1,064 data points), displayed a moderate correlation with the estimated population served (R = 0.21). In contrast to our results, a previous study in Kentucky found no significant correlation between the crAssphage wastewater concentration and the population served [22]. A study in Italy also found no correlation between crAssphage concentrations and the flow capacity of the wastewater treatment plant, as well as reporting that the wastewater concentration method used will significantly influence the quantified crAssphage concentration in wastewater [30]. While only employed in Colorado (20 sewersheds, 1,914 data points) the correlation between F+ Coliphage and the estimated population served was extremely low (R = 0.08). A previous study from Colorado that analyzed F+ Coliphage concentration in their wastewater samples reported that the concentrations also displayed inconsistent geographic and temporal trends among their study sites, suggesting that F+ Coliphage was not an effective population biomarker [28]. Overall, our study results do not support the use of either one of these three fecal indicator wastewater concentrations as reliable population biomarkers for WBE.

We then proceeded to assess flow rate as a potential population biomarker. We observed a very strong correlation between the sewershed-level flow rate (L/Day) and estimated population served across all study sites, with the strongest correlation coefficient in Wisconsin (R = 0.96) and the lowest correlation observed in Virginia (R = 0.76) (Fig 1). Notably, the lower correlation observed in Virginia is still much stronger than any of the correlations of the fecal indicator concentration with the estimated population served as described above.

Because of the strong correlation between flow rate and the estimated population served, we wanted to determine if flow normalizing the fecal indicator concentration would generate a stronger correlation with population than what we observed from using raw fecal indicator concentrations alone. Flow normalization of the fecal indicator concentration did improve the correlation with the estimated population served, but the overall correlation was still lower than that observed using flow rate by itself. The strongest correlation with the flow normalized fecal indicator concentration and the estimated population served was observed with PMMoV (R = 0.89) in Wisconsin. In Ohio and Virginia, the flow rate improved the correlation of PMMoV and the estimated population served, producing a moderate, and significant positive correlation. The flow normalized crAssphage concentration in Ohio produced a strong correlation with the population served (R = 0.62). A previous study by Wilder et al. conducting WBE of SARS-CoV-2 in New York state compared flow normalized crAssphage wastewater concentrations to the estimated population served and reported a correlation of 0.72 [29]. A previous study by Holm et al. discussed significant temporal and spatial variability of raw fecal indicator wastewater concentrations and concluded that flow normalization of the fecal

indicator concentration is an important step while conducting WBE [22]. While the results from our study show that flow normalization does improve the correlation between the fecal indicator and population served; flow rate by itself (without a fecal normalization) displayed the strongest correlation with the estimated population served.

A limitation of this analysis is that some areas will utilize combined sewer overflows (CSO), which take in wastewater, runoff, and precipitation, which could significantly increase the flow rate without an association to the population size. In our study we were not able to stratify our data by such variables because they were not consistently reported by each study site. Regardless, the results from our study still support the hypothesis that flow rate is strongly associated with the population served by an individual sewershed. Another limitation of our study is that we only investigated three of the CDC NWSS recommended population biomarkers. Many studies across the globe have proposed and investigated other population biomarkers such as β-2 microglobulin, creatinine, 5-hydroxyindoleacetic acid, caffeine, paraxanthine, ammoniacal total nitrogen, and total phosphorous [20,21,51]. These endogenous population biomarkers display a low to strong correlation with the population served (R = 0.06–0.8) [21]. Future multi-site studies across vast geographic and temporal space, such as this study, could include other population biomarkers along with flow rate to investigate and identify the most effective WBE population biomarker. Until such studies become available, we propose flow rate is the most consistent population normalization factor for WBE. Flow rate has the added advantage in that it is routinely measured by WWTP personnel in the U.S. and thus, can easily be included for population analysis when conducting WBE [52]. For communities where flow rate is not routinely measured, Maal-Bared et al. evaluated the cost-effectiveness of flow meters and suggest that it would be a helpful investment for facilitating future WBE efforts [47].

## Correlation of SARS-CoV-2 wastewater concentrations and COVID-19 clinical cases

The use of population normalization factors with measured SARS-CoV-2 wastewater concentrations is common for WBE; however there is not yet a consensus as to which normalization factor is the most effective for monitoring disease trends in a community. We assessed the correlation of the SARS-CoV-2 wastewater concentration with reported COVID-19 cases using the current CDC NWSS recommended normalization approaches [26]. The Spearman correlation coefficients with COVID-19 cases from our four chosen normalization categories of the SARS-CoV-2 wastewater concentrations ranged from -0.09 (Ohio) to 0.9 (California) (Fig 2). Among the correlation values of all normalization categories, California observed the strongest mean correlation (R = 0.76) between the SARS-CoV-2 wastewater concentrations and COVID-19 cases. Ohio observed the lowest mean correlation (R = 0.06) between the SARS-CoV-2 wastewater concentrations and COVID-19 cases among all four normalization categories. Negative correlations were observed in Ohio, and we attribute this to the poor N1 and N2 wastewater concentrations measured that we discussed above. The remaining study sites had similar mean correlations with their wastewater concentrations and COVID-19 cases (Fig 2). A T-Test of the overall correlations stratified by the CDC N1 and N2 targets showed that there was no significant difference between the results produced by either genetic target (Mean difference = -0.07, P-value = 0.343).

The correlation of the SARS-CoV-2 wastewater concentrations and COVID-19 clinical cases by each normalization category are displayed in Fig 3. The individual correlation values for each study site and normalization category can be found in S3 Table. Apart from the results we observed from Ohio, the correlation values reported among the study sites are consistent with previous findings, where the correlations with COVID-19 clinical cases typically ranged

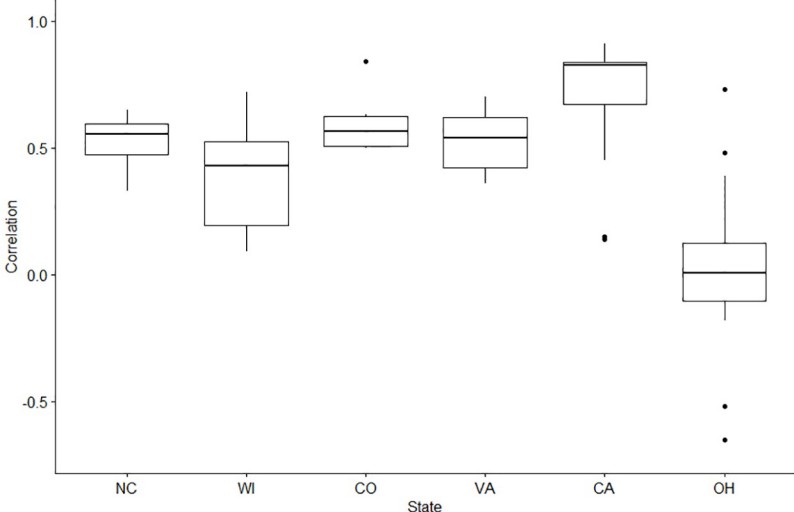

**Fig 2. Correlation coefficients of the SARS-CoV-2 wastewater concentrations and the sewershed-level COVID-19 cases for each study site.** This range includes the correlation coefficient using each of the four normalization categories used in the analysis.

from 0.5–0.9 [10,12,14,23,53]. A limitation of our study is that we were only able to assess three CDC NWSS recommended normalization factors. While we were not able to assess other normalization factors directly, the results from our study are similar to a range of correlation values of normalized SARS-CoV-2 wastewater concentrations and COVID-19 cases (R = 0.2–0.7) produced from studies that utilized chemical normalization factors such as β-2 microglobulin, caffeine, creatinine, paraxanthine, 5-hydroxyindoleacetic acid, total nitrogen, or total phosphorus [21,47,51,54,55]. We further stress the importance of additional assessments of other population biomarkers in community wastewater to identify other potential targets to further improve WBE of infectious diseases.

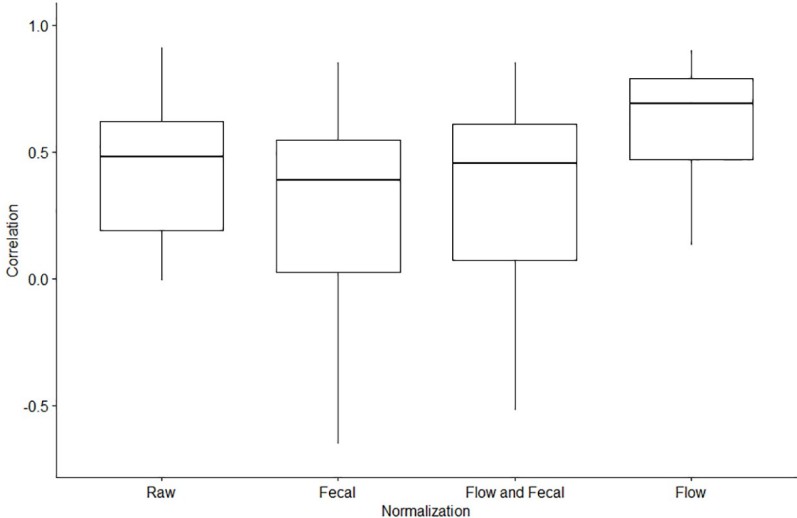

**Fig 3. Correlation coefficients of the SARS-CoV-2 wastewater concentrations and the sewershed-level COVID-19 cases for each study site by each normalization category.**

**Table 4. T-Test analysis of the difference of the correlation coefficients observed among each of the normalization categories.**

| Normalization Category | Mean Difference | P-Value |
|---|---|---|
| Flow–Raw | 0.16 | 0.02 |
| Flow—Fecal | 0.23 | 0.007 |
| Flow—Flow & Fecal | 0.19 | 0.022 |
| Fecal–Flow & Fecal | -0.04 | 0.022 |
| Fecal–Raw | -0.07 | 0.28 |
| Flow & Fecal—Raw | -0.03 | 0.559 |

Our assessment showed that flow normalized SARS-CoV-2 wastewater concentrations generated the strongest correlation with COVID-19 clinical cases. By contrast, the fecal normalized SARS-CoV-2 wastewater concentrations generated the weakest correlation with COVID-19 clinical cases. Our T-Test results showed that the flow normalized correlation coefficients were significantly higher than the other three normalization categories (Table 4). Previous studies have also reported strong correlations of flow normalized SARS-CoV-2 wastewater concentrations with COVID-19 clinical cases [11,51,53]. We also found that the flow and fecal normalized correlations were significantly higher than just the fecal normalized correlations (Table 4). In contrast to our results, a study in Canada reported that normalization did not significantly improve their correlation with COVID-19 cases, when compared to the correlation using the raw SARS-CoV-2 wastewater concentration [47]. Additionally, weak correlations of fecal normalizing SARS-CoV-2 wastewater concentrations with PMMoV and F + Coliphage have also been previously reported [14,20,21,56]. One study in California assessed the correlation from four normalization biomarkers (PMMoV, crAssphage, Bacteroides rRNA, and human 18S), and found that the biomarkers had a negative effect on the correlation with COVID-19 clinical cases [57]. A study in Wisconsin also reported a negative effect of fecal normalization on the overall correlation, similar to what we found across all of our included study sites [23]. A study from Colorado that used the F+ Coliphage for fecal normalization also reported that it did not improve results for trend analysis [28]. F+ Coliphage was subsequently dropped from the routine surveillance protocol and replaced with PMMoV in this study. Two separate studies from Ohio examined the effect of normalization factors on the correlation between SARS-CoV-2 wastewater concentrations and COVID-19 clinical cases. Nagarkar et al. reported that normalizing for flow rate or fecal indicators had no significant improvement of the overall correlation from any of the normalization factors [58]. Interestingly, the other Ohio study from Ai et al. reported a negative effect of fecal normalization on the correlation with COVID-19 clinical cases, similar to what we observed in our study [56]. The negative correlation values on the fecal normalized SARS-CoV-2 wastewater concentrations reported in ours and other studies, support the conclusion that fecal normalization may not provide a strong positive effect on disease trend analysis for WBE.

The overall results in our study showed that while flow normalized wastewater concentrations produced the strongest correlation with cases, there were instances within our study where the fecal normalized concentrations had the strongest correlation with COVID-19 cases. One example is in North Carolina, where the PMMoV normalized wastewater concentrations were higher than the flow normalized correlations, but the fecal and flow normalized correlations in North Carolina were the highest (S2 Table). This result in North Carolina suggests that, while flow normalized was not the strongest, it still improved the correlation in addition to fecal normalization. Similar to our observation, the potential enhancement of using fecal indicators in addition to flow normalized SARS-CoV-2 wastewater concentrations

was previously reported by Langeveld et al. [25]. Additionally, a study from Duvallet et al. analyzing nationwide data in the U.S. also reported that fecal normalization with PMMoV improved the correlation with COVID-19 cases in some study sites but it was lower than the correlation in other study sites when compared to the raw wastewater concentration [59]. Additionally, in our study the overall correlation values of the raw SARS-CoV-2 wastewater concentration and reported COVID-19 cases was not significantly different than the fecal normalized correlation values, the raw concentration had a smaller range that did not go as far into the negative correlation values. If flow rate is not available, it may be preferable to use the raw SARS-CoV-2 wastewater concentration rather than pursue a fecal normalization approach [47]. Our results show that flow normalized SARS-CoV-2 wastewater concentrations generate the strongest correlation with COVID-19 cases.

## Molecular analysis method influence on the correlation with COVID-19 cases

The optimal molecular analysis tool to be used for WBE is a topic of growing importance that has warranted further investigation on the effects of the different tools on results generated when processing wastewater samples [54,60–62]. This dataset allowed us to study the effect of laboratory methods and stratify the correlations generated from all the normalization categories by the molecular analysis approach used, i.e., RT-ddPCR and rRT-qPCR. Wastewater samples analyzed by RT-ddPCR generated significantly higher correlation values (Mean difference = 0.21, P-value = 0.008) than wastewater samples analyzed by rRT-qPCR (Fig 4). Our results suggest that molecular analysis by RT-ddPCR is more precise than rRT-qPCR, and that wastewater samples analyzed by RT-ddPCR will generate the best trends of COVID-19 cases within a community. Previous studies have also suggested RT-ddPCR as the preferred molecular analysis tool for wastewater surveillance due to its ability to produce more precise wastewater concentration results [60–62]. While RT-ddPCR is more expensive than rRT-qPCR, it is the optimal molecular analysis option, and it should become the standard approach used globally as the cost becomes more affordable to laboratories conducting WBE. Our assessment results conclude that utilizing RT-ddPCR and the flow normalization approach will produce

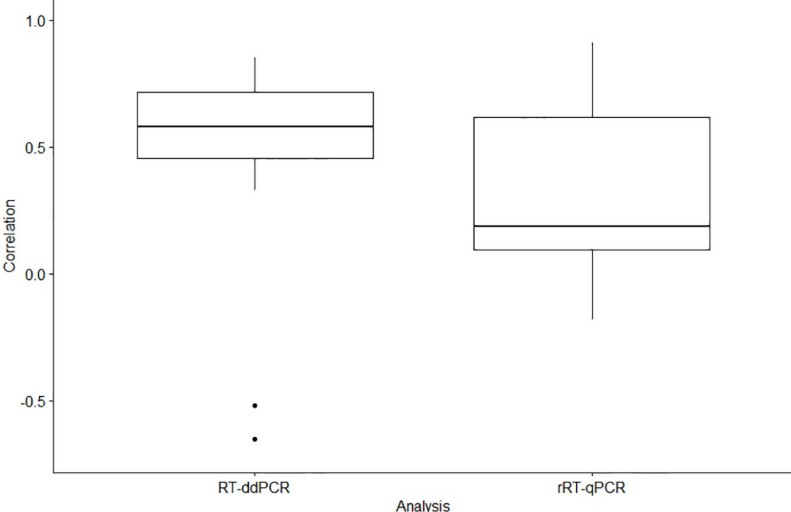

**Fig 4. Correlation coefficients of the SARS-CoV-2 wastewater concentrations and the sewershed-level COVID-19 cases for each study site by each molecular analysis method.**

the strongest COVID-19 community disease trend results which can be used to improve pandemic response across the globe. Consistent inter-laboratory processing procedures for WBE are important and can greatly improve generalization and interpretation of results [41]. We believe that a national standard operating procedure for processing wastewater samples should be prioritized in the future.

## Conclusions

Our study has provided the first multistate assessment of population normalization factors for WBE of SARS-CoV-2 using the CDC NWSS dataset. The dataset of 12,445 wastewater surveillance observations across 182 communities located throughout six U.S. states provided depth, breadth, and robustness to our study. With this dataset we assessed the CDC NWSS recommended normalization factors and determined which factors were accurate population biomarkers and the effect these normalization factors have on the correlation of SARS-CoV-2 wastewater concentration trends with reported COVID-19 cases. We found that the flow rate was the best population biomarker, having the strongest correlation with the estimated population served across all study sites. We also found that normalization of the SARS-CoV-2 wastewater concentration by flow rate produced the strongest correlation with reported COVID-19 cases. Lastly, our study revealed that wastewater samples analyzed using RT-ddPCR produced a stronger correlation with COVID-19 cases than samples that were analyzed using rRT-qPCR. Overall, our assessment has demonstrated that WBE of SARS-CoV-2 is an effective approach for monitoring trends of disease within a community, and that generated results can be enhanced by analyzing wastewater samples with RT-ddPCR and normalizing the wastewater concentration by the flow rate.

## Supporting information

**S1 Table. Influent wastewater processing methods used by each study site concentration, extraction, and molecular analysis of SARS-CoV-2 RNA.**
(DOCX)

**S2 Table. Spearman correlation coefficient analysis of the fecal indicator raw wastewater concentration (Log10GC/L), fecal indicator flow normalized wastewater concentration (Log10GC/Day), and flow rate (L/Day) with the estimated population served across each study site.**
(DOCX)

**S3 Table. Spearman correlation coefficients (P-Value) of each normalization category for the SARS-CoV-2 wastewater concentrations and the sewershed-level COVID-19 cases for each study site.**
(DOCX)

## Acknowledgments

We would like to thank the CDC NWSS team for their generous help in processing our data request and answering technical questions about the primary data input process. Centers for Disease Control and Prevention/Agency for Toxic Substances and Drug Registry, National Wastewater Surveillance System Restricted Data Set, 2021, as compiled from data provided through National Wastewater Surveillance System in the Waterborne Disease Prevention Branch.

## Author Contributions

**Conceptualization:** Andrew L. Rainey, Song Liang, Joseph H. Bisesi, Jr., Anthony T. Maurelli.

**Data curation:** Andrew L. Rainey.

**Formal analysis:** Andrew L. Rainey, Song Liang, Anthony T. Maurelli.

**Funding acquisition:** Andrew L. Rainey, Anthony T. Maurelli.

**Investigation:** Andrew L. Rainey, Song Liang, Anthony T. Maurelli.

**Methodology:** Andrew L. Rainey, Song Liang, Anthony T. Maurelli.

**Project administration:** Song Liang, Joseph H. Bisesi, Jr., Anthony T. Maurelli.

**Resources:** Andrew L. Rainey, Song Liang, Joseph H. Bisesi, Jr., Anthony T. Maurelli.

**Supervision:** Song Liang, Joseph H. Bisesi, Jr., Anthony T. Maurelli.

**Validation:** Song Liang, Anthony T. Maurelli.

**Visualization:** Andrew L. Rainey, Song Liang, Anthony T. Maurelli.

**Writing – original draft:** Andrew L. Rainey, Song Liang, Joseph H. Bisesi, Jr., Tara Sabo-Attwood, Anthony T. Maurelli.

**Writing – review & editing:** Andrew L. Rainey, Song Liang, Joseph H. Bisesi, Jr., Tara Sabo-Attwood, Anthony T. Maurelli.

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
