## [Decision Letter · Decision Letter 0]

21 Mar 2023

PONE-D-23-03823A multistate assessment of population normalization factors for wastewater-based epidemiology of COVID-19PLOS ONE

Dear Dr. Maurelli,

Thank you for submitting your manuscript to PLOS ONE. After careful consideration, we feel that it has merit but does not fully meet PLOS ONE’s publication criteria as it currently stands. Therefore, we invite you to submit a revised version of the manuscript that addresses the points raised during the review process. The manuscript has been carefully evaluated by two  experts in the field and some suggestions  have been made  in order to increase the quality of your paper. I do agree with these indications and consequently I invite you to undertake this small revision process in order to improve your manuscript.

We look forward to receiving your revised manuscript.

Kind regards,

Vittorio Sambri, M.D., Ph.D.

Academic Editor

PLOS ONE

Journal Requirements:

Reviewers' comments:

**Comments to the Author**

1. Is the manuscript technically sound, and do the data support the conclusions?

Reviewer #1: Yes

Reviewer #2: Yes

2. Has the statistical analysis been performed appropriately and rigorously? 

Reviewer #1: Yes

Reviewer #2: Yes

3. Have the authors made all data underlying the findings in their manuscript fully available?

Reviewer #1: Yes

Reviewer #2: Yes

4. Is the manuscript presented in an intelligible fashion and written in standard English?

Reviewer #1: Yes

Reviewer #2: Yes

5. Review Comments to the Author

Reviewer #1: The paper face a very important problem for Wastewater Based Epidemiology. The used methods are appropriate and well explained, the literature updated.

Some minor points to clarify:

Did the sewershed population vary along the time, for example due to the summer holidays?

Are cases reported as resident or present in the studied areas at the time of sampling?

The length of the sewerage was not taken in account for fecal normalization: could it affect the results increasing the variability of fecal indicators load?

Reviewer #2: The article titled: A multistate assessment of population normalization factors for wastewater-based

epidemiology of COVID-19" is a good, recent and very useful work. I recommend it for publication in your journal in a present form.

6. PLOS authors have the option to publish the peer review history of their article (what does this mean?). If published, this will include your full peer review and any attached files.

Reviewer #1: **Yes: **Annalaura Carducci

Reviewer #2: **Yes: **Kamila Zdenkova

---

## [Author Response · Author response to Decision Letter 0]

23 Mar 2023

Editor’s Comments

Formatting edits have been made throughout the revised manuscript document.

This has been completed. 

3. In your Data Availability statement, you have not specified where the minimal data set underlying the results described in your manuscript can be found. PLOS defines a study's minimal data set as the underlying data used to reach the conclusions drawn in the manuscript and any additional data required to replicate the reported study findings in their entirety. All PLOS journals require that the minimal data set be made fully available.

CDC NWSS data that was used for this study is publicly available and can be found at the following link: https://data.cdc.gov/Public-Health-Surveillance/NWSS-Public-SARS-CoV-2-Wastewater-Metric-Data/2ew6-ywp6

Reviewer #1: The paper face a very important problem for Wastewater Based Epidemiology. The used methods are appropriate and well explained, the literature updated.

1. Did the sewershed population vary along the time, for example due to the summer holidays?

The reported population for each sewershed did not vary in this dataset and was reported constantly with the same population served value for all data points. This was noted in the track changed manuscript (Lines 257-260).

2. Are cases reported as resident or present in the studied areas at the time of sampling?

CDC NWSS provided COVID-19 case data that was primarily collected and uploaded by each U.S. state. According to CDC, this data undergoes an additional quality check procedure to ensure accuracy of COVID-19 reporting (https://www.cdc.gov/coronavirus/2019-ncov/covid-data/faq-surveillance.html). Our assumption throughout this study was that reported cases were for individuals that are residents within each key sewershed and presumptively present in that key sewershed geographic area during the duration of infection.

3. The length of the sewerage was not taken in account for fecal normalization: could it affect the results increasing the variability of fecal indicators load?

While sewage travel time is a variable in our CDC NWSS dataset, it was only reported in 11% of the total data set, and only in two of the six study states (California and Ohio). Even among these two states, sewage travel time was not reported for all key sewersheds within each state. Currently, reporting states are not required to provide data for all variables of interest by CDC NWSS prior to upload for public use, which is why the majority of the data set has missing values for travel time. We did consider including sewage travel time when beginning this project. However, due to the above-mentioned limitations with the dataset, we decided not to include this variable in our study analysis. 

2. Reviewer #2: The article titled: A multistate assessment of population normalization factors for wastewater-based epidemiology of COVID-19" is a good, recent and very useful work. I recommend it for publication in your journal in a present form. 

This reviewer had no additional comments.

---

## [Editor Report · Decision Letter 1]

29 Mar 2023

A multistate assessment of population normalization factors for wastewater-based epidemiology of COVID-19

PONE-D-23-03823R1

Dear Dr. Maurelli,

We’re pleased to inform you that your manuscript has been judged scientifically suitable for publication and will be formally accepted for publication once it meets all outstanding technical requirements.

Kind regards,

Vittorio Sambri, M.D., Ph.D.

Academic Editor

PLOS ONE

---

## [Editor Report · Acceptance letter]

3 Apr 2023

PONE-D-23-03823R1 

A multistate assessment of population normalization factors for wastewater-based epidemiology of COVID-19 

Dear Dr. Maurelli:

I'm pleased to inform you that your manuscript has been deemed suitable for publication in PLOS ONE. Congratulations! Your manuscript is now with our production department. 

Kind regards, 

on behalf of

Professor Vittorio Sambri 

Academic Editor

PLOS ONE